# E3 ligases RNF43 and ZNRF3 display differential specificity for endocytosis of Frizzled receptors

Jeroen M Bugter, Peter van Kerkhof*, Ingrid Jordens*, Eline Janssen, Thi Tran Ngoc Minh, Daniel Iglesias van Montfort, Cara Jamieson , Madelon M Maurice

The transmembrane E3 ligases RNF43 and ZNRF3 perform key tumour suppressor roles by inducing endocytosis of members of the Frizzled (FZD) family, the primary receptors for WNT. Loss-of-function mutations in *RNF43* and *ZNRF3* mediate FZD stabilisation and a WNT-hypersensitive growth state in various cancer types. Strikingly, *RNF43* and *ZNRF3* mutations are differentially distributed across cancer types, raising questions about their functional redundancy. Here, we compare the efficacy of RNF43 and ZNRF3 of targeting different FZDs for endocytosis. We find that RNF43 preferentially down-regulates FZD1/FZD5/FZD7, whereas ZNRF3 displays a preference towards FZD6. We show that the RNF43 transmembrane domain (TMD) is a key molecular determinant for inducing FZD5 endocytosis. Furthermore, a TMD swap between RNF43 and ZNRF3 re-directs their preference for FZD5 down-regulation. We conclude that RNF43 and ZNRF3 preferentially down-regulate specific FZDs, in part by a TMD-dependent mechanism. In accordance, tissue-specific expression patterns of FZD homologues correlate with the incidence of *RNF43* or *ZNRF3* cancer mutations in those tissues. Consequently, our data point to druggable vulnerabilities of specific FZD receptors in *RNF43*- or *ZNRF3*-mutant human cancers.

## Introduction

The closely related transmembrane E3 ligases RNF43 and ZNRF3 control cell surface abundance of Frizzled (FZD) proteins, the primary receptors for WNT, by promoting their ubiquitination and lysosomal degradation (Hao et al, 2012; Koo et al, 2012). Because *RNF43* and *ZNRF3* both represent bona fide WNT target genes, the encoded proteins constitute a negative feedback regulatory role within the WNT cascade (Koo et al, 2012; Takahashi et al, 2014). Within the stem cell niche, this negative feedback role is inactivated by high local concentrations of RSPO proteins, to allow for the strong potentiation of WNT signalling required for the maintenance of adult stem cell populations (Hao et al, 2012).

In the mouse small intestine, deletion of either *Rnf43* or *Znrf3* did not induce a discernible phenotype, whereas the combined deletion of both genes resulted in the expansion of the stem cell zone, and eventually adenoma formation (Koo et al, 2012). The essential roles of Rnf43 and Znrf3 in adult stem cell homeostasis were shown for multiple tissues, including the liver, adrenal gland, and tongue (Basham et al, 2019; Belenguer et al, 2022; Lu et al, 2022). Of note, whereas both E3 ligases show redundant expression in certain tissues, such as the intestine, in other tissues homeostasis depends on expression of only one of the two E3 ligases. Interestingly, these observations correlate with the differential prevalence of *RNF43* and *ZNRF3* mutations in human cancer subtypes originating from specific tissues (Bugter et al, 2021). The functional relevance of this differential mutational pattern of both E3 ligases remains unclear.

WNT signalling regulation at the cell surface is complex, illustrated by the encoding of 19 WNT genes, 10 FZD genes and various co-receptor genes within the human genome (Niehrs, 2012). Downstream WNT-FZD signalling can be divided into β-catenin-dependent (canonical) and β-catenin-independent (non-canonical) pathways. Certain WNTs or FZD receptors are reported to activate either canonical or non-canonical WNT signalling, whereas others can activate both pathways depending on the recruitment of co-receptors (Niehrs, 2012). Various studies indicate that RNF43 and ZNRF3 are capable of inhibiting both β-catenin-dependent and -independent WNT pathways (Hao et al, 2012; Moffat et al, 2014; Tsukiyama et al, 2015; Radaszkiewicz et al, 2021). The mechanism by which RNF43/ZNRF3 interact with members of the FZD family,

Oncode Institute and Centre for Molecular Medicine, UMC Utrecht, Utrecht, Netherlands

Correspondence: m.m.maurice@umcutrecht.nl
Jeroen M Bugter's present address is Institute of Molecular Oncology and Functional Genomics, Center for Translational Cancer Research (TranslaTUM), School of Medicine, Technische Universität München, Munich, Germany
Eline Janssen's present address is Department of Medical BioSciences, Radboud university medical center, Nijmegen, Netherlands
Thi Tran Ngoc Minh's present address is Division Cell Biology, Metabolism and Cancer, Department Biomolecular Health Sciences, Faculty of Veterinary Medicine and Biomolecular Mass Spectrometry and Proteomics, Bijvoet Centre for Biomolecular Research, Utrecht University, Utrecht, Netherlands
Daniel Iglesias van Montfort's present address is Hubrecht Institute, Royal Netherlands Academy for Arts and Sciences, University Medical Center Utrecht, Utrecht, Netherlands
*Peter van Kerkhof and Ingrid Jordens contributed equally to this work

however, has remained a debated issue. Whereas one study proposed that the extracellular protease-associated (PA) domain of RNF43 interacts with the cysteine-rich domain of FZD (Tsukiyama et al, 2015), others reported that the RNF43 PA domain is dispensable for WNT signalling suppression (Radaszkiewicz et al, 2020). In another model, intracellular bridging of RNF43 and FZD is mediated by the scaffold protein Dishevelled (DVL) (Jiang et al, 2015), but also these findings have remained disputed because of the observation that RNF43 may interact with FZD in a manner independent of the RNF43-DVL interaction (Tsukiyama et al, 2015, 2020). Importantly, both models do not allow for the identification of receptor specificity. Thus, the precise mechanism of RNF43/ZNRF3-mediated FZD recognition and down-regulation remains unaddressed.

WNT hypersensitivity has emerged as a major driver of cancer growth, which may be caused either by mutational inactivation of *RNF43*/*ZNRF3* or overexpression of RSPO-fusion proteins (Koo et al, 2012; Seshagiri et al, 2012). These observations have led to a growing interest in the application of upstream WNT signalling inhibitors as an anti-cancer strategy (Bugter et al, 2021). As a result, several pan-FZD inhibitors and WNT-secretion inhibitors are currently being evaluated in clinical trials. Although conclusive clinical efficacy data are still lacking, these strategies are accompanied by side effects caused by a lack of selectivity and/ or tissue specificity (Jung & Park, 2020; Zhong & Virshup, 2020). For several cancer subtypes, however, cancer growth was shown to rely on one specific FZD homologue, which holds promise for more selective inhibition strategies (Steinhart et al, 2017; Do et al, 2022).

The level of functional redundancy of RNF43- and ZNRF3-mediated FZD targeting has remained unclear. Over recent years, several studies reported unique roles of post-translational regulation of one of the two E3 ligases, or unique roles and activities beyond WNT signalling (Ci et al, 2018; Spit et al, 2020; Kim et al, 2021; Fang et al, 2022). Furthermore, the observation that several cancer types display specific loss of only one of the two E3 ligases suggests that non-redundant functions exist. In this study, we uncover the specific roles of RNF43 and ZNRF3 in the down-regulation of FZD receptors. In addition, by examining tissue-specific expression patterns we translate FZD-specific roles to the incidence of *RNF43* and *ZNRF3* mutations in cancer.

# Results and Discussion

### RNF43 and ZNRF3 suppress WNT3A- and WNT5A-mediated signalling

To examine the activity of overexpressed RNF43 or ZNRF3 variants and avoid the effects of endogenous protein expression, we generated *RNF43*/*ZNRF3* double-knockout HEK293T cells (R/ZdKO). Similar to previous reports (Jiang et al, 2015; Radaszkiewicz et al, 2020; Spit et al, 2020), R/ZdKO cells displayed a strong increase in basal WNT/β-catenin signalling activity in comparison with WT HEK293T cells, which was further enhanced by WNT3A stimulation, as shown by a WNT luciferase reporter assay (Fig 1A). As expected, R/ZdKO cells were unresponsive to RSPO1 treatment (Fig 1A) (Jiang

et al, 2015; Radaszkiewicz et al, 2020). For evaluation of β-catenin-independent WNT signalling induced by WNT5a (Yamaguchi et al, 1999; González-Sancho et al, 2004), we examined phosphorylation levels of DVL2, an immediate downstream effector of FZD-mediated signalling that acts in both β-catenin-dependent and -independent WNT pathways (González-Sancho et al, 2004; Bryja et al, 2007). We blocked interference of endogenous WNT ligands using WNT-secretion inhibitor C59. R/ZdKO cells displayed increased levels of DVL2 phosphorylation upon WNT3A or WNT5A stimulation in comparison with WT cells (Fig 1B). Co-treatment with RSPO1 further increased the levels of DVL2 phosphorylation of WT cells, whereas R/ZdKO cells were insensitive to RSPO1 treatment (Fig 1B). Overexpression of either RNF43 or ZNRF3 in R/ZdKO cells reduced levels of DVL2 phosphorylation upon WNT3A or WNT5A treatment (Fig 1C). In line with previous findings, these results suggest that both RNF43 and ZNRF3 possess the capacity to inhibit both β-catenin-dependent and -independent WNT pathways (Hao et al, 2012; Moffat et al, 2014; Tsukiyama et al, 2015; Radaszkiewicz et al, 2021).

### RNF43 and ZNRF3 preferentially target specific FZD receptors for endocytosis

Based on their differential tissue expression and mutational patterns (Bugter et al, 2021), we wondered if RNF43 and ZNRF3 may display a preference for targeting specific FZDs for endocytosis (Hao et al, 2012; Koo et al, 2012). To investigate this issue, we used a set of expression constructs for nine human FZDs (Do et al, 2022) and examined their sensitivity for ligase-induced down-regulation from the plasma membrane.

For most FZDs, except for FZD4, we observed a clear distinction between higher and lower molecular weight (MW) bands on Western blot (Fig 2A). For FZD5, we previously determined that the higher MW band represents the mature, post-Golgi glycosylated form, whereas the lower band represents an immature high mannose-modified ER-localised form (Gerlach et al, 2018). Using cell surface biotinylation and streptavidin pull-downs, we here confirmed that most FZD members displayed two MW forms, with the higher molecular weight form representing the mature, plasma membrane-localised fraction of the protein (Fig S1A). FZD8 showed relatively low overall expression levels and, consequently, the mature protein forms were detectable only after immunoprecipitation (Fig 2A). In addition, FZD3 did not express at all and was therefore excluded from further analysis.

We co-expressed each FZD with RNF43 or ZNRF3 in R/ZdKO cells and examined the capacity of the two E3 ligases to degrade individual FZD members, by calculating the mature-to-immature ratio of FZD by Western blotting (Figs 2A and B and S1B). Whereas most FZDs were down-regulated by both E3 ligases the degree of down-regulation was variable, ranging from very efficient (e.g., FZD8) to (almost) no down-regulation (e.g., FZD9) (Fig 2B). Strikingly, when comparing the down-regulation efficiency of both E3 ligases we noticed that RNF43 most potently down-regulated FZD1, FZD5, and FZD7, whereas ZNRF3 displayed a small but significant preference for FZD6 (Fig 2B).

To further examine E3 ligase specificity for these FZD subgroups, we co-expressed RNF43 or ZNRF3 with FZD5 or FZD6 containing an

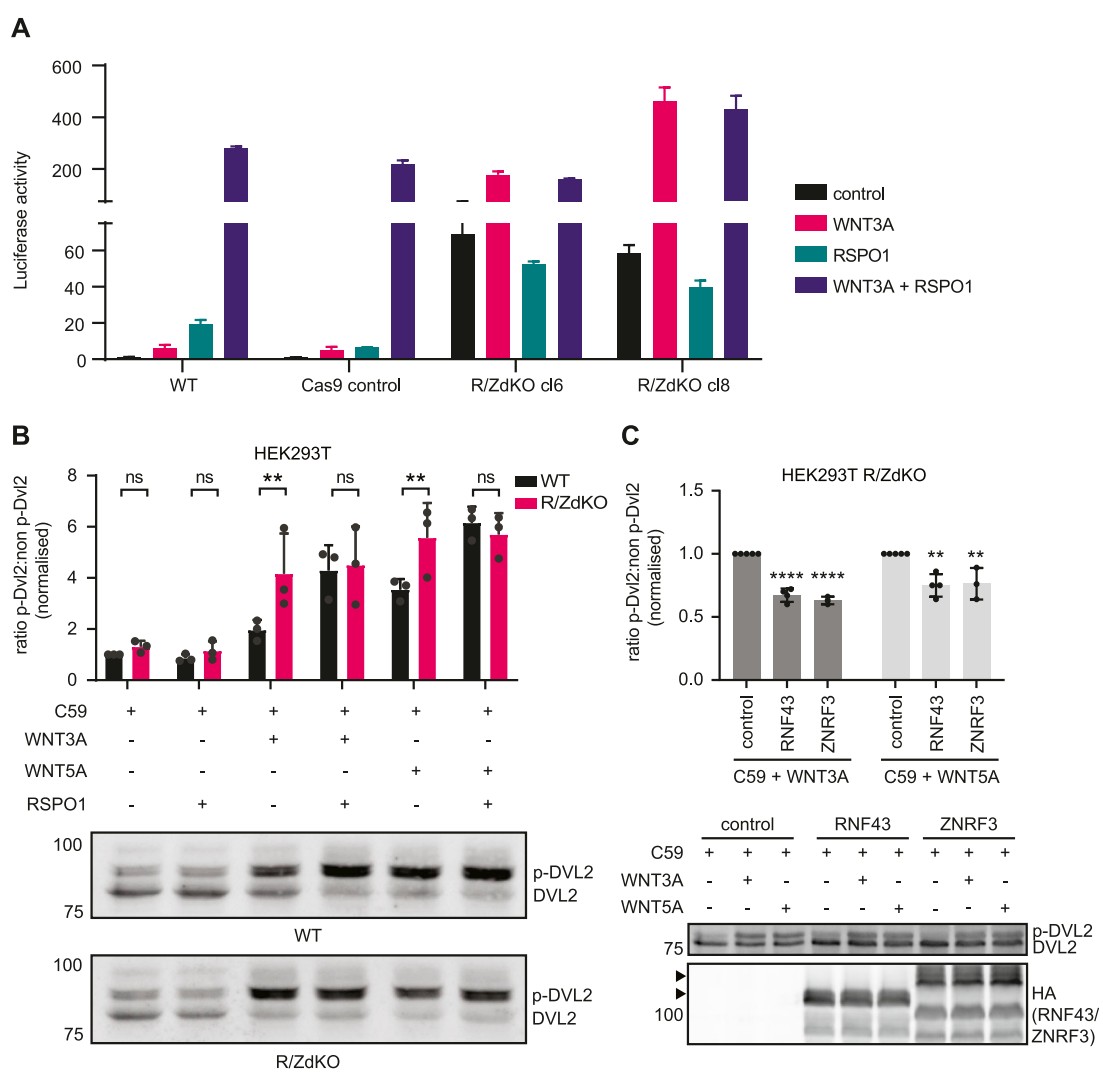

**Figure 1. RNF43 and ZNRF3-mediated suppression of WNT3A- and WNT5A-induced signalling.**
**(A)** WNT/β-catenin–mediated luciferase reporter assay performed with WT HEK293T cells, HEK293T Cas9-expressing control cells or two clonal HEK293T RNF43/ZNRF3 double-knockout (R/ZdKO) cell lines that were overnight stimulated with WNT3A and/or RSPO1-conditioned medium (CM) compared with control medium. Average β-catenin–mediated reporter activities + SD. in n = 2 independent wells are shown for a representative experiment. (N = 2). **(B)** Western blot analysis of DVL2 phosphorylation in WT HEK293T and R/ZdKO cells after 3 h stimulation with recombinant WNT3A or WNT5A and/or RSPO1-CM. All samples were pretreated with porcupine inhibitor (C59; 5 µM). Ratio between phosphorylated DVL2 (p-DVL2) and non-phosphorylated DVL2 protein band density was calculated, normalised to C59 control, and plotted. A representative experiment (N = 3) is shown. **(C)** Western blot analysis of DVL2 phosphorylation in HEK293T R/ZdKO cells overexpressing RNF43, ZNRF3, or control vector after 3 h stimulation with recombinant WNT3A or WNT5A. All samples were pretreated with porcupine inhibitor (C59; 5 µM). Ratios between phosphorylated and non-phosphorylate DVL2 were normalised to cells expressing control vector and plotted. A representative experiment (N = 2–4) is shown. Arrowheads indicate full-length ZNRF3 and RNF43. **(B, C)** Data information: data are presented as mean + SD. ns, not significant. ** ($P < 0.01$), **** ($P < 0.0001$). (B): Multiple $t$ test with multiple comparison correction, (C): one-way ANOVA.
Source data are available for this figure.

extracellular SNAP-tag and performed a pulse chase experiment to determine the level of ligase-mediated FZD internalisation, by labelling with a cell-impermeable fluorescent SNAP probe (Koo et al, 2012). Similar to our previous findings (Koo et al, 2012), overexpression of RNF43 induced a rapid reduction in the FZD5 fraction that resides at the cell surface (Figs 2C and D and S2A and B). By contrast, ZNRF3 caused only a moderate reduction in the surface fraction of FZD5 whereas consistently inducing slightly more efficient internalisation of FZD6 in comparison with RNF43 (Fig 2C and D). In an alternative approach, we quantified FZD5 and FZD6

surface levels upon co-expression of both E3 ligases using flow cytometry. These experiments again confirmed a preferential down-regulation of FZD5 by RNF43, and FZD6 by ZNRF3 (Figs 2E and F and S2C and D). Of note, co-expression of LRP6, a previously suggested substrate for ZNRF3 (Hao et al, 2012), did not affect the differential preference of RNF43 and ZNRF3 for FZD5 down-regulation (Fig S2E). Taken together, although both RNF43 and ZNRF3 are capable of suppressing WNT signalling via FZD down-regulation, each homologue appears to possess a ligase-specific substrate preference.

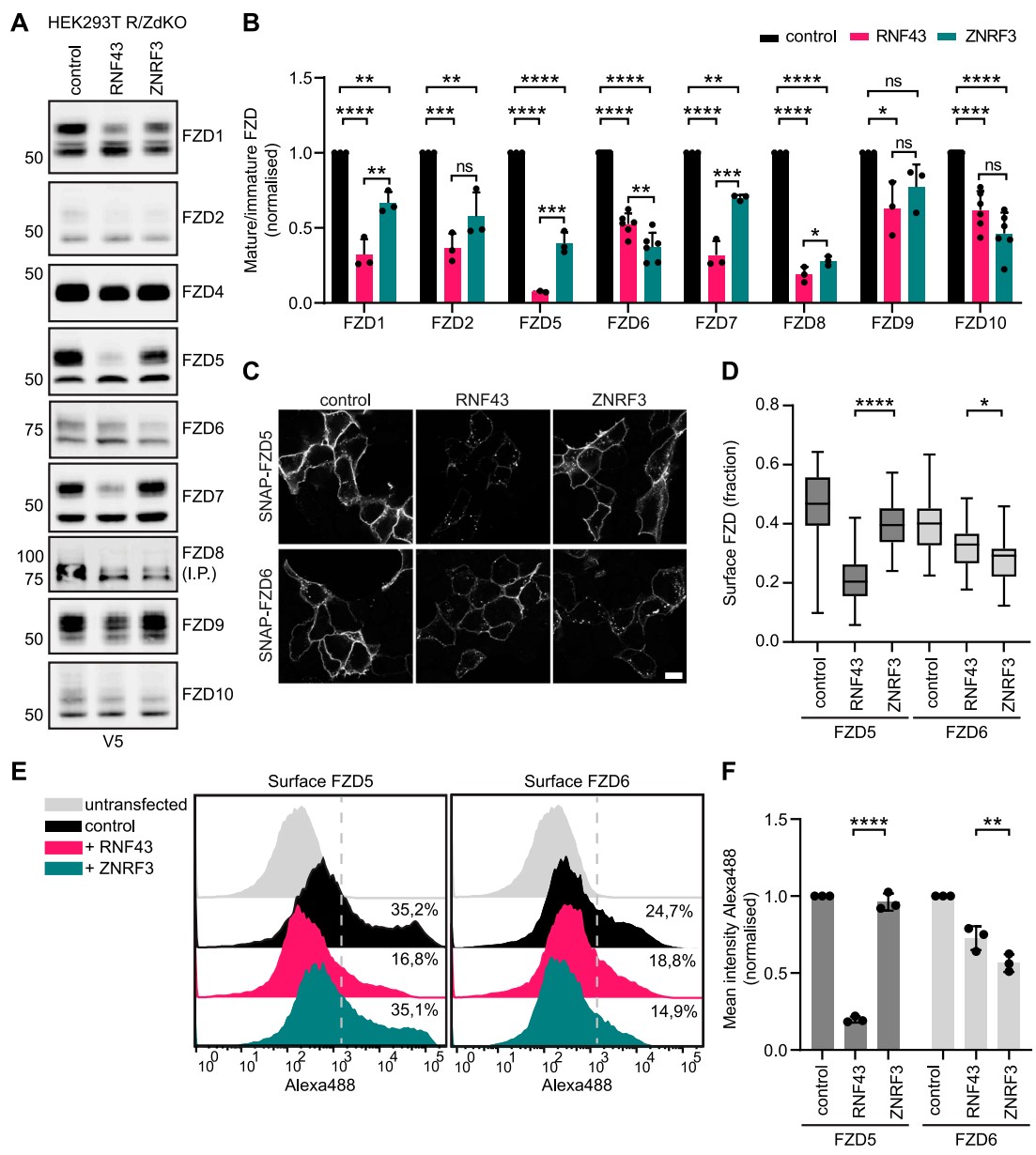

**Figure 2. RNF43 and ZNRF3 display Frizzled substrate specificity.**
**(A)** Western blot analysis of V5-tagged human FZD constructs upon co-expression with RNF, ZNRF3, or control vector. V5-immunoprecipitated (IP) samples are blotted for FZD8, total cell lysate for the other FZDs. **(B)** Relative levels of mature FZD are shown as the ratio between mature and immature FZD forms, normalised to pcDNA3-transfected control. (N = 3–6). **(C)** Immunofluorescence showing the subcellular localisation of SNAP-FZD5 or SNAP-FZD6 co-transfected with RNF43, ZNRF3 or control vector. Surface SNAP-FZD was labelled for 15 min and chased for either 30 (FZD5) or 60 min (FZD6). Scale bar, 10 $\mu$m. **(C, D)** Quantification of surface FZD fractions shown in (C). Graph shows the average surface fractions for the indicated conditions (N = 33–58 cells). **(E)** Representative FACS plots showing anti-V5-Alexa488 staining of non-permeabilized cells expressing V5–FZD5 or V5-FZD6 co-transfected with RNF43, ZNRF3, or control vector. Percentage of cells positive for surface FZD is determined by gating the Alexa488 signal based on untransfected cells. **(F)** Quantification of mean Alexa488 intensities of the experiment shown in (E) normalised to the control (N = 3).
**(B, D, F)** Data information: data are presented as mean + SD. ns, not significant. * ($P < 0.05$), ** ($P < 0.01$), *** ($P < 0.001$), and **** ($P < 0.0001$) (one-way ANOVA). Source data are available for this figure.

## Molecular requirements for RNF43-mediated FZD down-regulation

To determine the protein regions that confer substrate specificity of RNF43 and ZNRF3, we first replaced their endogenous signal peptides by the highly efficient H2-Kb signal peptide commonly used for optimized expression of type I membrane proteins

(Bénaroch et al, 1995; Fiebiger et al, 2005). Replacement of the signal peptide did not affect the capacity of RNF43 and ZNRF3 to differentially down-regulate FZD5 (Fig 3A), indicating that substrate specificity must be dictated by different regions within these E3 ligases.

We next investigated which regions of the RNF43 protein are required for FZD5 down-regulation, by generating various RNF43

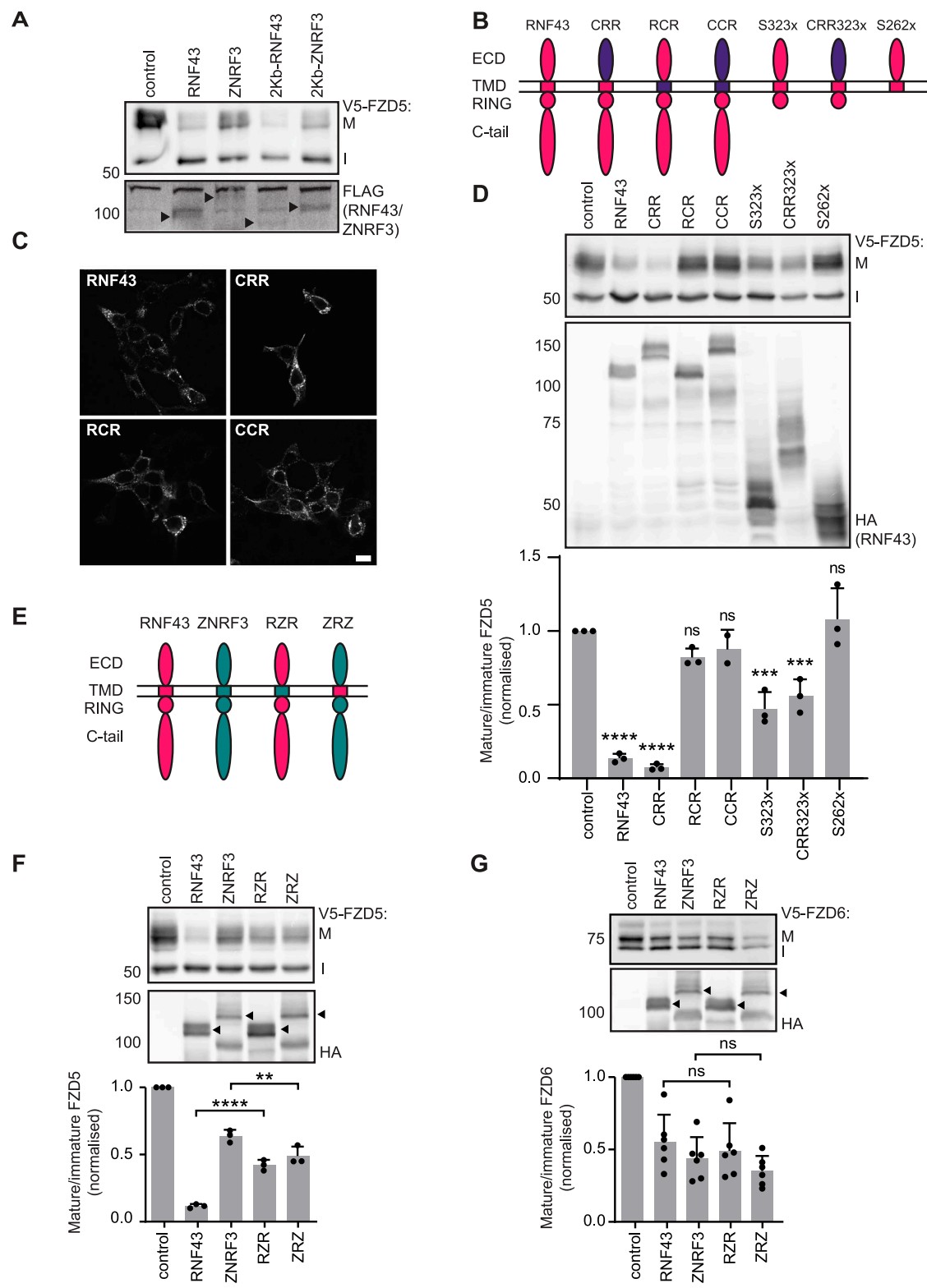

**Figure 3. The RNF43 transmembrane domain (TMD) is essential for FZD5 down-regulation and determines FZD specificity.**
**(A)** FZD5 down-regulation capacity of RNF43 and ZNRF3 with their endogenous signal peptides (2XFLAG, 2XHA) or a generic H-2Kb signal peptide (1XFLAG). Mature (M) and immature (I) FZD5 are indicated. Full-length RNF43 and ZNRF3 are indicated with black arrowheads. **(B)** Schematic representation of the RNF43 (pink) and CD16/7 (purple) chimeric and truncated constructs used in this study. Extracellular domain (ECD), transmembrane domain (TMD), E3 ligase RING domain (RING), C-terminal tail (C-tail). **(C)** Immunofluorescence showing the subcellular localisation of RNF43, CRR, RCR, and CCR in HEK293T cells. Scale bar, 10 μm. **(D)** FZD5 down-regulation capacity of RNF43 constructs with various protein domains replaced or truncated. Relative levels of mature FZD5 are shown as the ratio between mature (M) and immature (I) forms,

deletion and chimeric variants. We generated RNF43 constructs in which we replaced the extracellular domain (ECD) with that of the unrelated protein CD16 (CRR), swapped the transmembrane domain (TMD) for the similarly sized TMD of CD7 (RCR) or replaced both regions (CRR) (Fig 3B). We confirmed that the subcellular local-isation of each variant was comparable with WT RNF43 (Fig 3C). Functionally, we found that replacement of RNF43-ECD with CD16-ECD (CRR) did not affect the capacity of RNF43 to down-regulate FZD5 (Fig 3D). These findings indicate that RNF43-ECD is dispensable for FZD5 down-regulation and suppression of WNT/β-catenin signalling, in agreement with previous findings (Radaszkiewicz et al, 2020), but in disagreement with another report (Tsukiyama et al, 2015). Strikingly, replacement of RNF43-TMD with CD7-TMD (RCR) significantly diminished the capacity of RNF43 to down-regulate FZD5 (Fig 3D). These findings thus reveal a previously unknown role of RNF43-TMD for efficient FZD5 down-regulation.

In an earlier report, we showed that a large fraction of the intracellular C-terminus of RNF43 is dispensable for its role in FZD5 down-regulation (Spit et al, 2020). To further map the functionally required region, we truncated the RNF43 C-terminus further, removing all known functional domains downstream of the RING domain (variant S323x; Fig 3D). Functional analysis revealed that RNF43 S323x variants are still capable of down-regulating FZD5, albeit with somewhat attenuated activity (Fig 3D). Notably, the RNF43 S323x truncant also lacks the previously identified DVL Interaction Region (DIR), which was proposed to be essential for the interaction of RNF43 with FZD (Jiang et al, 2015). Even a chimeric RNF43 variant carrying a CD16-derived ECD along with an early truncation downstream of the RING domain (CRR S323x) retained partial functionality in these assays (Fig 3D). Truncation upstream of the catalytic RING domain, however, completely abolished the FZD5 down-regulating capacity of RNF43 (variant S262x, Fig 3D). Thus, our results indicate that the RNF43-TMD and RING domain comprise the minimally required fraction of the protein for FZD5 down-regulation, with a minor role of the C-terminal tail.

### The TMD determines RNF43 and ZNRF3 substrate specificity

We next wondered whether substrate specificity of RNF43 and ZNRF3 is dictated by their TMD. To address this issue, we swapped the TMDs of the two E3 ligases (Fig 3E) and tested their efficacy in down-regulating FZD5 and FZD6. Strikingly, ZNRF3 containing the RNF43 TMD (ZRZ) showed an increased FZD5 down-regulation ef-ficacy compared with WT ZNRF3, whereas RNF43 containing the ZNRF3 TMD (RZR) showed a decreased efficacy in down-regulating FZD5 compared with WT RNF43 (Fig 3F). By contrast, TMD swapping did not alter the efficacy of either RNF43 or ZNRF3 in down-regulating FZD6 (Fig 3G). Importantly, swapping of TMDs did not affect the overall intracellular localisation of both E3 ligase variants

(Fig S2F). These data indicate that the substrate specificity of these closely related E3 ligase proteins is dictated, at least partly, by their TMDs.

### Relation between tissue distribution of RNF43 and ZNRF3 cancer mutations and Frizzled expression

To assess the biological relevance of our findings, we examined the distribution of FZD expression across various human tissue types. Remarkably, FZD expression patterns displayed a sub-stantial level of tissue specificity. Whereas some tissues showed higher expression of RNF43-preferred substrates (FZD1, 5, 7), others mainly expressed ZNRF3-preferred substrate FZD6 (Fig 4A). FZD5 is the dominant homologue in the pancreas, colon, small intestine, and liver. By contrast, FZD6 is the dominant homologue in the adrenal gland and skin (Fig 4A). Notably, cancer types arising from the latter two organs are enriched for mutations in *ZNRF3* in comparison with mutations in *RNF43* (Fig 4B). In line with these observations, *Znrf3* loss, but not *Rnf43* loss, is sufficient to transform adrenal cortical tissue in mouse models (Fig 4C) (Basham et al, 2019), although both E3 ligases are expressed in these tissues (Fig S3). By contrast, *RNF43* mutations, but not *ZNRF3* mutations, are enriched in pancreatic ductal adenocarcinoma (PDAC) (Fig 4B), and *Rnf43* loss promotes PDAC formation in *Kras*-mutant mouse models (Fig 4C) (Hosein et al, 2022). Accordingly, *RNF43*-mutant PDAC cell lines and xenografts appear exclusively dependent on signalling via FZD5 (Steinhart et al, 2017). Another example is provided by the critical role of Rnf43 but not Znrf3 in oligodendrocyte maturation upon injury (Niu et al, 2021). This process requires the down-regulation of FZD1 that we identified as an RNF43-preferred substrate. Lastly, in metastatic thyroid cancer, ZNRF3 was found to be commonly targeted by miR146b-5p resulting in FZD6 stabilisation at the cell surface and activation of WNT signalling and EMT (Deng et al, 2015).

In conclusion, we here reveal a substantial level of substrate specificity for the closely related E3 ligases RNF43 and ZNRF3 that were previously considered as functionally redundant (Koo et al, 2012). Furthermore, our results uncover a role of the TMD of these E3 ligases in determining substrate preference. In line with these findings, the amino acid sequence of the TMD is poorly conserved between RNF43 and ZNRF3, with only 1/21 (4, 7%) overlap in amino acid composition (Fig S2G). By contrast, the PA domain displays 46/100 (46%) and the RING domain 31/45 (69%) overlap in amino acids for both proteins (Tsukiyama et al, 2021). Regulation of FZD activity was shown previously to depend on TMD-mediated interactions. For instance, the TMD domain of TMEM59 was shown to promote clustering of FZD receptors (Gerlach et al, 2018), and dimerisation of FZD6 is dependent on homotypic TMD-TMD interactions (Petersen et al, 2017). Whether the TMDs of RNF43 and ZNRF3 are involved in direct intramembrane interactions with FZDs or rather affect their

---

normalised to pcDNA3-transfected control. (N = 2–3). **(E)** Schematic representation of RNF43 (pink), ZNRF3 (green) and TMD-swapped RZR and ZRZ constructs. **(F, G)** FZD5 (F) and FZD6 (G) down-regulation capacity of the constructs depicted in (E). Relative levels of mature FZD are shown as in (D) for indicated constructs (with or without swapped TMD). (N = 3–6). Arrowheads indicate full-length ZNRF3 and RNF43. **(D, F, G)** Data information: data are presented as mean + SD. ns, not significant. ** (*P* < 0.01), *** (*P* < 0.001), and **** (*P* < 0.0001) (one-way ANOVA).
Source data are available for this figure.

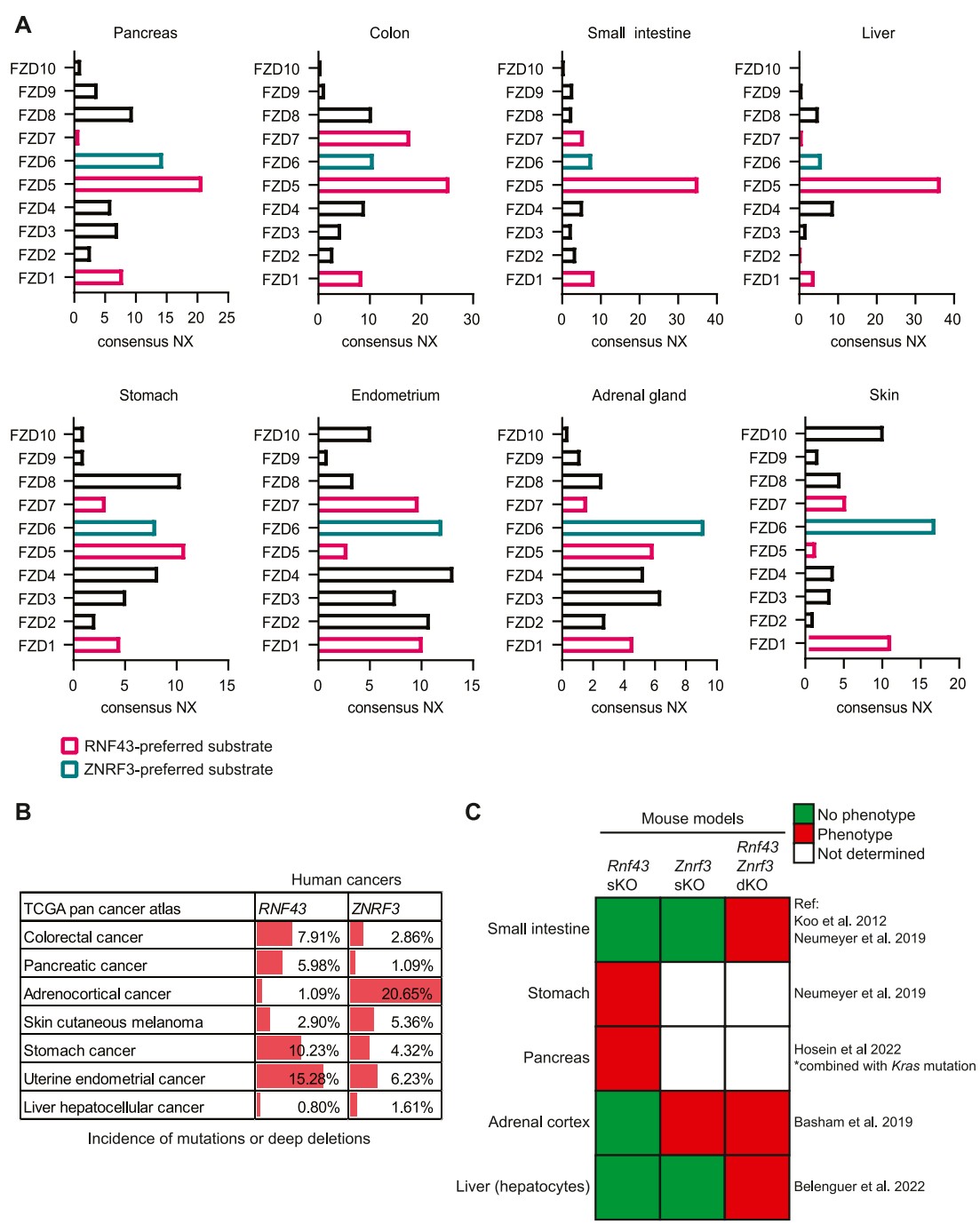

**Figure 4. Frizzled homologue expression in various human tissues relates to incidence of RNF43 and/or ZNRF3 mutations in cancer or phenotypes in mouse models.**
**(A)** Consensus normalised expression (NX) of all FZD genes among human tissues. RNF43- and ZNRF3-preferred substrates are indicated in pink and green, respectively. **(B)** Incidence of *RNF43* and *ZNRF3* mutations and deep deletions among the cancer genomics atlas pan cancer atlas datasets. **(C)** Results of published mouse studies indicated reported phenotype of *Rnf43*, *Znrf3* single-knockout (sKO), or double-knockout (dKO) (Koo et al, 2012; Basham et al, 2019; Neumeyer et al, 2021; Belenguer et al, 2022; Hosein et al, 2022).

localisation to specific membrane domains remain open questions for further investigation. As the transmembrane swap did not affect the FZD6 down-regulating capacity of ZNRF3, it is possible that the precise mechanism varies between the different ligase-receptor pairs. Overall, our findings provide an explanation for the differential incidence of tissue-specific mutation frequency of *RNF43* and *ZNRF3* in cancers. Understanding the preferences of these highly mutated E3 ligases might help in predicting vulnerabilities to inhibition of specific FZD receptors in *RNF43*- or *ZNRF3*-mutant cancers.

## Materials and Methods

### Cell culture and generation of knockout lines

Human embryonic kidney (HEK) 293T cells were cultured in DMEM high glucose (Invitrogen), supplemented with 10% FBS (GE Health-Care), 2 mM UltraGlutamine (Lonza), 100 units/ml penicillin, and 100 µg/ml streptomycin (Invitrogen). Cells were cultured at 37°C in 5% $CO_2$. WNT3a-conditioned medium (CM) and RSPO1-CM were produced as described (Tauriello et al, 2010; Fenderico et al, 2019).

RNF43 and ZNRF3 double-knockout cells (R/ZdKO) were generated by simultaneously targeting *RNF43* (gRNA: ATTGCACAGGTA-CAGCGGGT) and *ZNRF3* (gRNA: GCCAAGCGAGCAGTACAGCG) with gRNAs cloned in a pSpCas9(BB)-2A-Puro vector (# 48139; Addgene). Monoclonal cell lines that were homozygous knockout for both genes were confirmed by genotyping and functional analysis in a *β*-catenin-mediated reporter assay (Fig 1A).

### Plasmids and transfection

RNF43–2xFlag–HA (FFHH) and ZNRF3-FFHH constructs were described previously (Koo et al, 2012). V5-mouse FZD5 was described earlier (Tauriello et al, 2012) and V5-human FZD6 was generated similarly. RNF43 mutants were generated by PCR-subcloning using Q5 High-Fidelity 2× Master Mix (NEB). Domain-swapped constructs were generated by in-fusion cloning (Takara Bio). Human FZD(1-10)-V5-IRES-mKate constructs were a kind gift of Karl Willert (Department of Cellular and Molecular Medicine, University of California San Diego, San Diego, USA). Human myc-LRP6 was a gift from Christoph Niehrs (Institute of Molecular Biology, Mainz, Germany). All constructs were sequence verified. Transfections were performed using either FuGENE6 (Promega) according to manufacturer's protocol for *β*-catenin-mediated reporter assays and microscopy or polyethylenimine (PEI) for Western blot and flow cytometry analysis.

### β-catenin-mediated reporter assays

HEK293T WT or R/ZdKO cells were seeded in 24-well plates and transfected the next day with 30 ng of reporter construct TOPFlash or FOPFlash, 5 ng of thymidine kinase (TK)-Renilla and the indicated constructs. Cells were stimulated 6 h post-transfection with WNT3a-CM and/or RSPO1-CM overnight, then cells were lysed in passive lysis buffer (Promega) for 20 min at RT. Levels of Firefly and Renilla luciferase were measured using the dual-luciferase kit (Promega) according to the manufacturer's instructions on a Berthold luminometer Centro LB960.

### WNT3A and WNT5A response assays

Response to canonical and non-canonical WNT ligands, respectively, WNT3A and WNT5A, was measured as previously reported (Radaszkiewicz et al, 2021). HEK293T R/ZdKO cells were treated with 5 µM PORCN inhibitor C59 (Tocris) to block secretion of endogenous WNT ligands. Cells were transfected with RNF43-FFHH, ZNRF3-FFHH or pcDNA3 control vector and incubated with 5 µM C59 overnight. The next day, cells were stimulated with 100 ng/ml recombinant WNT3A (5036-WN; R&D Systems) or WNT5A (645-WN; R&D Systems). Cells were lysed in lysis buffer (1% Triton-X100 + 150 mM NaCl + 1 mM EDTA in 20 mM Tris pH 7,5 + 50 mM NaF + 1 mM PMSF + 10 µg/ml aprotinin + 10 µg/ml leupeptin + 1 mM $Na_3VO_4$) and boiled in 1x SDS sample buffer for 5 min.

### FZD down-regulation assays

HEK293T R/ZdKO cells were co-transfected with V5-tagged FZD and RNF43-FFHH, ZNRF3-FFHH or pcDNA3 control vector. The next day, cells were subjected to cell surface biotinylation (see below) or directly lysed in lysis buffer (1% Triton-X100 + 150 mM NaCl + 1 mM EDTA in 20 mM Tris pH 7,5 + 50 mM NaF + 1 mM PMSF + 10 µg/ml aprotinin + 10 µg/ml leupeptin + 1 mM $Na_3VO_4$). In case of FZD8, because of low expression, immunoprecipitation was used to concentrate protein levels (see below). In the other cases, total protein lysates were eluted in SDS sample buffer and incubated for 30 min at 37°C.

### FZD immunoprecipitation

Protein lysates were incubated with 1 µg mouse anti-V5 antibody for 2 h at 4°C, followed by 1 h incubation with 25 µl Protein A-Agarose Beads (RepliGen). Beads were washed five times with 0,1 x PBS after which beads were eluted in SDS sample buffer and incubated for 30 min at 37°C.

### Cell surface biotinylation

Live cells were washed three times with ice-cold PBS complete (including $Ca^{++}$ and $Mg^{++}$) and incubated for 30 min on ice in PBS complete with 0.8 mM Sulfo-NHS-SS-Biotin (Thermo Fisher Scientific). Cells were washed once with PBS complete, once with DMEM medium + 50 mM Glycine to quench unreacted biotin for 5 min on ice and once with PBS complete. Cells were collected and lysed in lysis buffer. Protein lysates were incubated with 25 µl Streptavidin-agarose beads (Pierce, Thermo Fisher Scientific) for 1 h at 4°C. Beads were washed five times with 0,1 x PBS after which beads were eluted in SDS sample buffer and incubated for 30 min at 37°C.

### Western blotting and analysis

Western blotting was performed using standard procedures. Samples were resolved by SDS–PAGE, transferred to Immobilon-FL PVDF membranes (Millipore, Sigma-Aldrich), blocked with Odyssey blocking buffer (Li-Cor), incubated with the indicated primary antibodies overnight at 4°C, with secondary antibodies for 1 h at RT, and imaged using the Amersham Typhoon NIR laser scanner (GE HealthCare). Quantifications were performed using ImageQuant TL 8.2 (GE HealthCare). Activation of downstream signalling in the WNT3A and WNT5A response assays was determined by calculating the ratio of phosphorylated over non-phosphorylated DVL2 bands on Western blot (Radaszkiewicz et al, 2021). FZD down-regulation was determined by calculating the ratio of the mature over the immature band, except for FZD4 for which the two isoforms were indistinguishable and the total levels were calculated instead.

### Immunofluorescence and SNAP surface labelling

HEK293T cells were seeded on laminin-coated glass coverslips. The next day cells were transfected with indicated constructs. 20 h post-transfection, cells were fixed in 4% paraformaldehyde for 30 min and subsequently the reaction was quenched by incubation with 0.05 M NH₄Cl for 30 min at RT. Cells were blocked in blocking buffer (PBS with 2% BSA [Sigma-Aldrich] and 0.1% saponin). Next, cells were incubated with primary and secondary antibodies diluted in blocking buffer for 1 h at RT. Cells were mounted in ProLong Gold (Life Technologies) and analyzed using a Zeiss LSM700 confocal microscope.

For SNAP surface labelling, cells were incubated with 1 $\mu$M SNAP-Alexa488 (Bioke) for 15 min at RT. Subsequently, cells were chased for 30 or 60 min at 37°C and washed once with warm medium before fixation and staining (see above).

### Quantification surface fraction SNAP-FZD

Confocal images were processed in ImageJ (1.53) to determine the surface FZD fractions. To measure the total intensity of FZD per cell, a region of interest (ROI) was drawn lining the outside of the cell. Subsequently, by drawing a ROI right below the cell surface, the intracellular intensity of FZD signal was measured. Next, the surface fraction was calculated by subtracting the intracellular intensity from the total intensity. Next, the fraction of surface levels over total was calculated. The ROI for total FZD was transferred to the second channel to determine the total level of RNF43 or ZNRF3 by anti-Flag labelling (Fig S2A and B).

### Flow cytometry

To measure surface levels of V5-FZD5 or -FZD6, HEK293T R/ZdKO cells were seeded in six wells plates and transfected the next day

with V5-FZD5 or V5-FZD6 and either control vector, RNF43-FFHH or ZNRF3-FFHH using PEI. 24 h after transfection, cells were detached in PBS containing 5 mM EDTA for 5 min at 37°C and collected by centrifugation for 5 min at 300$g$ at 4°C. Cells were washed twice with PBS containing 0.5% BSA and blocked in PBS containing 1% BSA for 30 min on ice. Subsequently, cells were stained with mouse anti-V5 for 30 min and washed three times with PBS containing 0.5% BSA at 4°C. Next, cells were stained with goat anti-mouse Alexa488 in PBS containing 1% BSA for 30 min on ice. Afterwards, cells were washed three times with PBS containing 0.5% BSA at 4°C and resuspended in 250 $\mu$l PBS containing 0.5% BSA and 1 $\mu$M Sytox-blue (Invitrogen) for live-dead staining. Data were analyzed using FlowJo v10 (BD Biosciences). Fluorescence of a minimum of 5,000 singlet-gated cells was measured by flow cytometry on a BD Celesta Flow Cytometer (BD Biosciences) using a FITC 488 nm laser for Alexa488 and a BV421 nm laser for Sytox-Blue.

### Public data analysis

To address FZD expression across human tissues, normalised consensus RNA expression data (NX) were downloaded from the Human Protein Atlas version 22.0. The frequency of *RNF43* and *ZNRF3* mutations in human cancer subtypes was addressed in The Cancer Genomics Atlas (TCGA) Pan Cancer Atlas data that were accessed via Memorial Sloan Kettering Cancer Center (MSKCC) cBioPortal. Mutations and deep deletions were combined.

### Statistical analysis

Statistical analysis on Western blot, IF, and FACS quantification data was performed in GraphPad Prism 9.5.1 using One-Way ANOVA with a Tukey's multiple comparison test to calculate *P*-values for

**Antibodies.**

| Name | Manufacturer and catalogue number | Dilution for Western blot | Dilution for immunofluorescence/flow cytometry |
|---|---|---|---|
| Rabbit anti-DVL2 | 3216; 3224; Cell Signaling | 1:1,000 | X |
| Rabbit anti-V5 | V8137; Sigma-Aldrich | 1:2,000 | X |
| Mouse anti-Flag | F3165; Sigma-Aldrich | 1:5,000 | 1:1,000 |
| Mouse anti-V5 | A01724; Genscript | 1:5,000 | 1:500 |
| Rabbit anti-Flag | F7425; Sigma-Aldrich | 1:2,500 | 1:500 |
| Rat anti-HA | 11867423001; Roche | 1:1,500 | x |
| Rabbit anti-myc | 06-549; Sigma-Aldrich | 1:2,000 | x |
| Mouse anti-actin | 691001; MP | 1:10,000 | x |
| Goat anti-rabbit IRDye680 | 926-32221; Li-Cor | 1:8,000 | X |
| Goat anti-mouse IRDye680 | 925-68070; Li-Cor | 1:8,000 | X |
| Goat anti-rabbit IRDye800 | 926-32211; Li-Cor | 1:8,000 | X |
| Goat anti-mouse IRDye800 | 926-32210; Li-Cor | 1:8,000 | X |
| Goat anti-rat Alexa680 | A21096; Invitrogen | 1:3,000 | x |
| Goat anti-rabbit Alexa488 | A11034; Invitrogen | X | 1:300 |
| Goat anti-mouse Alexa568 | A11031; Invitrogen | X | 1:300 |

comparison of individual groups. Multiple *t* tests with multiple comparison correction using the Holm-Sidak method were used in Fig 1B.

## Supplementary Information

## Acknowledgements

We thank Karl Willert (Department of Cellular and Molecular Medicine, University of California San Diego, San Diego, USA) for kindly providing the library of FZD(1-10)-V5-IRES-mKate constructs. We thank members of the laboratory of MM Maurice for discussions and suggestions. This work is part of the Oncode Institute, which is partly financed by the Dutch Cancer Society. This work was supported by the ZonMW TOP Grant 91218050 (to MM Maurice), Dutch Cancer Society grant 13112 (to MM Maurice), NWO Gravitation project IMAGINE! (to MM Maurice), and Dutch Cancer Society/TKI-Life Sciences and Health grant 2022-PPS-1/14853 (to MM Maurice).

### Author Contributions

JM Bugter: conceptualization, formal analysis, investigation, visualization, methodology, and writing—original draft, review, and editing.
P van Kerkhof: conceptualization, formal analysis, investigation, visualization, and methodology.
I Jordens: conceptualization, formal analysis, investigation, visualization, methodology, and writing—review and editing.
E Janssen: formal analysis, investigation, and methodology.
T Tran Ngoc Minh: formal analysis, investigation, and methodology.
D Iglesias van Montfort: formal analysis, investigation, and methodology.
C Jamieson: formal analysis, investigation, and methodology.
MM Maurice: formal analysis, supervision, funding acquisition, investigation, visualization, methodology, and writing—original draft, review, and editing.

### Conflict of Interest Statement

MM Maurice is an inventor on patents related to membrane protein degradation; she is co-founder and shareholder of Laigo Bio.

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
