## [Reviewer comments · Life Science Alliance]

Life Science Alliance

E3 ligases RNF43 and ZNRF3 display differential specificity for endocytosis of Frizzled receptors

Jeroen Bugter, Peter van Kerkhof, Ingrid Jordens, Eline Janssen, Thi Tran Ngoc Minh, Daniel Iglesias van Montfort, Cara Jamieson, and Madelon Maurice

DOI: <https://doi.org/10.26508/lsa.202402575>

Corresponding author(s): Madelon Maurice, University Medical Center Utrecht

Review Timeline:

Submission Date:	2024-01-05
Editorial Decision:	2024-02-02
Revision Received:	2024-06-05
Editorial Decision:	2024-06-24
Revision Received:	2024-06-25
Accepted:	2024-06-25

Transaction Report:

February 2, 2024

Re: Life Science Alliance manuscript #LSA-2024-02575-T

Prof. Madelon M. Maurice
University Medical Center Utrecht
Center for Molecular Medicine and Oncode Institute
Heidelberglaan 100
Utrecht 3584CX
Netherlands

Dear Dr. Maurice,

Thank you for submitting your manuscript entitled "E3 ligases RNF43 and ZNRF3 display differential specificity for endocytosis of Frizzled family members" to Life Science Alliance. The manuscript was assessed by expert reviewers, whose comments are appended to this letter. We invite you to submit a revised manuscript addressing the Reviewer comments.

Thank you for this interesting contribution to Life Science Alliance. We are looking forward to receiving your revised manuscript.

Sincerely,

B. MANUSCRIPT ORGANIZATION AND FORMATTING:

Reviewer #1 (Comments to the Authors (Required)):

Bugter et al. provide convincing evidence that the TM region of RNF43 provides specificity for its targeting of a subclass of FZD proteins (predominantly FZD1,5,7). This is a novel finding and will be of interest to the Wnt community, where understanding the selectivity of Wnt receptor biology is a major and important topic.

The experimental evidence provided in Fig2/3 is much stronger for RNF43 specificity towards FZD1,5,7, but this is not so clear for ZNRF3 targeting of FZD6/10. Although the data presented in Fig.4 tends to support their hypothesis that ZNRF3 preferentially targets FZD6/10 - because cancers in skin and adrenal gland that correlate more with ZNRF3 mutations also display high levels of FZD6/10 expression - I think the authors need to tone down their claims here.

One question comes to mind - does co-expression of LRP6 alter the selectivity? This could be done e.g., for FZD5. Have the authors looked into this?

The authors may think this is beyond the scope of the current ms, however I feel it is important information that could be gained by a relatively simple experiment.

The following should be clarified/corrected by the Authors:

Figure 2:

In Fig.2a the authors should comment on the results (from the perspective of WB protein band intensities) that for FZD8, and perhaps FZD10, both RNF43 and ZNRF3 have very similar activity, i.e., no apparent specificity. Also, for FZD10 there appears to be almost no activity directed by either RNF43 or ZNRF3.

For FZD4 how can the authors determine mature/immature band ratios in panel B, considering there only a single WB band visible in panel A? Does a lower exposure reveal two closely associated bands?

Comparing FZD5 with FZD6 makes sense considering they show opposing specificities for ZNRF3 (FZD5) and RNF43 (FZD6) (Figure 2a), however the weak expression levels for FZD6 makes it difficult to accurately quantify the ratio of mature/immature bands on WB and both bands appear equally reduced by ZNRF3. The additional IP concentration of FZD6 proteins likely also makes it more difficult to accurately determine.

Also, in Fig.2D, the reduction of cell surface FZD6 by ZNRF3 is not obvious, in contrast to the clearer reduction of cell surface FZD5 by RNF43.

This is not a criticism, however I feel that the lack of clearly robust data for FZD6 should be pointed out more.

In Fig.S2, it appears that there is a significant reduction in the expression of ZNRF3 when FZD6 is co-expressed - can the authors comment on this?

Figure 3:

Fig3B has a confusing construct nomenclature that needs clarification so that readers can understand quickly without having to read on for it to eventually become clear. Also remove the last 3 ZNRF3 constructs from 3B and add to 3E instead.

e.g.,

"We replaced either the extracellular domain (ECD) of RNF43 with the ECD of CD16, (CRR), its transmembrane domain (TMD) with the TMD of CD7 (RCR) as well as replacement of both (CCR) (Fig. 3B).

What about removal to the ECD of RNF43 completely, without substitution? This may help to clarify any role it plays...

I think the following sentence needs correction "Even a chimeric RNF43 variant carrying a CD16-derived ECD along with an

early truncation downstream of the RING domain (CRR323X) retained functionality in these assays, albeit with somewhat attenuated activity (Fig. 3D)."

In order to state "Thus, our results indicate that the RNF43-TMD and RING domain comprise the minimally required fraction of the protein for FZD5 downregulation", one would need to show activity for the Δ ECD/ Δ C-tail version of RNF43...

Referee Cross-Comments:

Both referees 2 and 3 are substantially more critical of the points that I raised regarding the data presented (especially in Figure 2). My own experience, however, would confirm that there is significant variation between the biochemistry of different FZD's and it is indeed challenging to have clearly comparable results. Nevertheless, I believe that the main conclusions of the authors are supported by the experimental data. I do however agree with some of the criticisms raised that more care needs to be taken when concluding e.g., that ZNRF3 shows specificity towards FZD6.

Overall, my opinion is that, although qualitatively sound and robust data are crucial, the "bigger picture" of a new finding is of equal significance if the data presented generally support the claims of the authors. Here, I believe this to be the case, although some improvements do indeed need to be made and more discussion on the variations between the different FZD's commented on and discussed in more detail.

More generally, in my opinion it is of more value to our field if we publish such preliminary findings more rapidly (as long as the study is sound). Often, extensive revisions may not, in the end, significantly change the overall quality or principal findings made. A slightly more tolerant approach may speed up the peer reviewing and publication process and should generally help as it will allow others to further refine the details once a new finding is published. In other disciplines, there appears to be a more forgiving/tolerant approach when it comes to peer review.

Reviewer #2 (Comments to the Authors (Required)):

The manuscript "E3 ligases RNF43 and ZNRF3 display differential specificity for endocytosis of Frizzled family members" by Madelon Maurice and co-authors aims at deciphering the substrate specificity, among the ten FZD proteins, of the two transmembrane E3 ligases RNF43 and ZNRF3. Studies on the domain(s) responsible for the substrate specificity, along with the correlations of these specificities and the relevance of the two ligases and their FZD substrates in cancer, are also provided. Overall, this is an interesting manuscript offering important and interesting insights. However, several issues question the validity of the authors' conclusions, and will need to be addressed in a major revision.

Major issues:

1. Two bands (or two groups of bands) of different FZDs visible in Westerns (Fig. 2A) are used by the authors to distinguish between the immature and mature forms of the receptors. The authors claim in the text: "Using cell surface biotinylation and streptavidin pull-downs, we confirmed that for all FZD members displaying two MW forms, the higher molecular weight form represents the mature, plasma membrane-localised fraction of the protein (Fig. S1)". This statement is false regarding FZD6 and FZD8, as they are not visible at all in the Fig. S1 due to the low expression levels, and the authors have to perform IPs for these receptors in their Fig. 2A. As FZD6 is a key isoform concluded in the rest of the paper to be the specific target of one of the two E3 ligases, this experimental deficit puts in question the validity of these subsequent conclusions regarding this FZD. Another problem relates to FZD4: the authors acknowledge that this is the isoform for which they do not see two separate bands. This, however, does not impede the authors to present the data on the ratio of the mature to immature bands for FZD4 in Fig. 2?

2. The whole concept of differential specificity of RNF43 vs. ZNRF3 to the nine tested FZDs is based on the fact that the upper-migrating band of these FZDs is differentially reduced upon expression of RNF43 or ZNRF3. This is the cornerstone of the paper, and in this regard the experimental issues raised in the point 1 above regarding FZD4, FZD6 and FZD8 become even more serious. However, there are more problems with these data presented in Fig. 2. Indeed, the Westerns in Fig. 2A are representatives of only 3 experiments, as stated in the legend. In this regard, the lack of statistical significance analysis for the quantifications of these Westerns presented in Fig. 2B is very troublesome. While the data for some receptors, such as FZD5 and perhaps FZD1 and FZD7, will likely be confirmed upon the statistical analysis, the differences presented for the other receptors are two modest. And with the low number of experiments these differences won't survive, I am afraid, the statistical assessment. Yet these miniscule differences are the basis for the authors to claim that ZNRF3 has the substrate specificity towards FZD6 (already highly troublesome, as discussed above) and FZD10.

In general, I find the way the data are presented in Fig. 2C misleading.

The data further provided in Fig. 2D, E also speak against one of the main conclusions of the authors, that ZNRF3 has the substrate specificity towards FZD6. In fact, the data cumulatively rather suggest (upon verification with the proper experimental additions and statistical assessment) that while RNF43 indeed has a preference towards some FZDs (such as FZD1, FZD5 and FZD7), ZNRF3 is unselective. Or the other way around, FZD1, FZD5 and FZD7 display the specificity towards RNF43 while the other FZDs can be more or less equally recognized by either of the two E3 ligases.

3. The interpretation of the experiments in Fig. 3A-D, where different truncation and replacement forms of RNF43 and ZNRF3

are tested for their capacity to downregulate FZD5, is made as if all the forms tested are present in equal amounts. However, these amounts are vastly different, as the anti-FLAG Western blots show, either due to their different expressions or stabilities. The data on the resulting FZD5 levels should be normalized to the levels of expression of the E3 ligase variants. Then the conclusions will be quite different from those reached by the authors. For example, the authors claim that "...replacement of RNF43-ECD with CD16-ECD (CRR) did not affect the capacity of RNF43 to downregulate FZD5..". However, CRR is expressed at the levels that look ca. 10-fold higher than the parental RNF43, thus, equal downregulations of FZD5 by the two constructs (by the way, the quantifications in Fig. 3D lack statistical significance analysis) may in reality reflect strongly decreased specific activity of CRR. Further, the authors write that removal of the intracellular part, which was by some others reported to interact with DVL and be important for the activity of RNF43, does not affect the activity of the E3 ligase to FZD5. However, this form (called R323x) is expressed at dramatically higher levels than the parental RNF43, thus the partial reduction in FZD5 levels this form induces, when normalized to the expression levels, will essentially give loss of the specific activity of this variant. When the authors treat their data properly, a less clear-cut picture will emerge: the TMD will indeed look crucial for the RNF43 activity towards FZD5, but the extracellular domain and the intracellular domains will also have a contributing importance, in agreement with some prior studies.

The same normalization to the expression levels will probably increase the strength of the authors' conclusions regarding the transmembrane domain-swapping experiments that create chimeras between RNF43 and ZNRF3. In Fig. 3E that investigates the downregulation of FZD5, the authors currently struggle with the fact that both chimeras they test (called RZR and ZRZ) have similar, intermediate levels of FZD5 downregulation. However, RZR has very high expression levels, thus its specific activity is quite reduced, while the specific activity of ZRZ will be higher given its low expression levels. The same logic then will apply to the FZD6 experiments in Fig. 3F.

Other issues:

1. The data provided in Fig. 1 need precisions regarding statistics. While multiple repeats are behind the panel (A) as evidenced by the error bars, the figure legend does not specify the number of experiments nor what the error bars are (sd or sem). For the panel (B), it is stated that a representative experiment of 4 independent experiments is shown. While this is fine for the Western blot shown, the quantification shown must be given as means \pm sd (or sem). Further, the statistical analysis of the significance of the differences presented must be provided (and its nature described in the legend). All these points also apply to the panel (C), where even the number of experiments is not stated, so currently it can be deduced that this panel represents a single experiment.
2. The nature of error bars in Fig. 2B, C is not specified. Analysis of statistical significance in the panel (B) must be performed and provided.
3. In Fig. 3A, the lower Western blot panel is supposed to show the levels of expression of FLAG-tagged RNF43 and ZNRF3 constructs, either with the endogenous signal peptides or with the H2-Kb signal peptides. However, the bands are hardly seen. And it is not at all clear why changing the endogenous signal peptide with the exogenous H2-Kb signal peptide (both should cleave off) should produce the proteins with unequal molecular weight.

Reviewer #3 (Comments to the Authors (Required)):

The paper from Bugter and colleagues examines the role of two E3 ligases, RNF43 and ZNRF3 in the endocytosis of the Wnt receptors Frizzled.

They report that loss of function of these enzymes results in the stability of Fz receptors at the cell surface resulting in increased sensitivity to Wnt ligands. RNF43 preferentially down regulates FZD1, 5 and 7 whereas ZNRF3 affects FZD6 and FZD10. The authors went on to examine the domains of these enzymes that affects this downregulation. They also examined the tissue expression of these two enzymes in relation to where FZD receptors are expressed. The data presented seems minimal and an incomplete story. In addition, the authors did not perform statistical analyses in many of the graphs presented. This is really surprising.

The manuscript has limited amount of data and the data is poorly presented. Therefore, this manuscript is not appropriate for Life Science Alliance.

Specific comments:

- 1) No statistical analyses are presented in Figure 1 B and C, Figure 2 D, E and F etc.
- 2) The figures are not properly labelled.
- 3) Figure 3 C, the images are not informative without quantification.

Rebuttal Bugter et al, Ms # LSA-2024-02575-T

Reviewer #1 (Comments to the Authors (Required)):

Bugter et al. provide convincing evidence that the TM region of RNF43 provides specificity for its targeting of a subclass of FZD proteins (predominantly FZD1,5,7). This is a novel finding and will be of interest to the Wnt community, where understanding the selectivity of Wnt receptor biology is a major and important topic.

The experimental evidence provided in Fig2/3 is much stronger for RNF43 specificity towards FZD1,5,7, but this is not so clear for ZNRF3 targeting of FZD6/10. Although the data presented in Fig.4 tends to support their hypothesis that ZNRF3 preferentially targets FZD6/10 - because cancers in skin and adrenal gland that correlate more with ZNRF3 mutations also display high levels of FZD6/10 expression - I think the authors need to tone down their claims here.

We thank the reviewer for their supportive comments. We agree that the preferential targeting of FZD6 by ZNRF3 is more subtle than the effects of RNF43, yet results were consistent across experiments and now validated using three different approaches (biochemistry, microscopy and flow cytometry). To better substantiate our findings, we now repeated all biochemical experiments for determining FZD6 downregulation using an optimized protocol that eliminated the need for IP (see below) and showed that the differential downregulation of FZD6 by RNF43 and ZNRF3 was statistically significant (**new Figure 2A and B**). As significance was not reached for FZD10, we adjusted our conclusions accordingly in the manuscript.

One question comes to mind - does co-expression of LRP6 alter the selectivity? This could be done e.g., for FZD5. Have the authors looked into this?

The authors may think this is beyond the scope of the current ms, however I feel it is important information that could be gained by a relatively simple experiment.

We thank the reviewer for this interesting suggestion. We performed the FZD5 downregulation experiment with and without co-expression of LRP6 for both E3 ligases and observed no differences in their targeting of FZD5 (**new Figure S2E**).

The following should be clarified/corrected by the Authors:

Figure 2:

In Fig.2a the authors should comment on the results (from the perspective of WB protein band intensities) that for FZD8, and perhaps FZD10, both RNF43 and ZNRF3 have very similar activity, i.e., no apparent specificity. Also, for FZD10 there appears to be almost no activity directed by either RNF43 or ZNRF3.

We agree with the reviewer and have adapted the description of our findings on page 6 lines 149 – 154.

For FZD4 how can the authors determine mature/immature band ratios in panel B, considering there only a single WB band visible in panel A? Does a lower exposure reveal two closely associated bands?

We apologize for the confusion. Indeed, for FZD4 we were not able to distinguish between mature and immature forms and we therefore quantified the total FZD fraction, which we explained in the methods section. For clarity, we removed FZD4 from main figure 2B and placed the quantification of total FZD4 levels in a separate graph in the supplemental figures (**new Figure S1B**).

Comparing FZD5 with FZD6 makes sense considering they show opposing specificities for ZNRF43 (FZD5) and RNF43 (FZD6) (Figure 2a), however the weak expression levels for FZD6 makes it difficult to accurately quantify the ratio of mature/immature bands on WB and both bands appear equally reduced by ZNRF3. The additional IP concentration of FZD6 proteins likely also makes it more difficult to accurately determine.

Also, in Fig.2D, the reduction of cell surface FZD6 by ZNRF3 is not obvious, in contrast to the clearer reduction of cell surface FZD5 by RNF43.

This is not a criticism, however I feel that the lack of clearly robust data for FZD6 should be pointed out more.

We understand the concerns of the reviewer and we agree that the differences in efficacy of RNF43 and ZNRF3 towards FZD6 are small, although consistently observed. We now substantiated these data in several ways. First, we optimized our method for FZD6 expression which eliminates the need for an IP step to visualize mature/immature ratios of FZD6. Using this method, we now repeated the experiments of **figures S1A, 2B, 3F** and **3G** and replaced blots and added quantifications of our new results. In addition, we repeated the microscopy experiments shown in Figure 2C-D, performed quantification of our results and added a statistical comparison of the effects of RNF43 and ZNRF3 (**new Figure 2D**). Finally, we added a FACS-based readout as a third method to quantify the effects of RNF43 and ZNRF3 on downregulating the cell surface pool of FZD6 (**new Figures 2E-F and S2C-D**). In all cases, the downregulation of FZD6 by ZNRF3 was significantly increased in comparison to RNF43. We added these results to the revised manuscript and we emphasize that the observed differences are small, but significantly different for the 2 E3 ligases (p6, line 153-154).

In Fig.S2, it appears that there is a significant reduction in the expression of ZNRF3 when FZD6 is co-expressed - can the authors comment on this?

We thank the reviewer for bringing up this point. We noticed that another replicate of the same experiment did not show a reduction in ZNRF3 upon FZD6 co-expression, making it unlikely that this is a biological effect. We now combined the quantification of two independent experiments in **new Figures 2D and S2B** to compensate for this experimental variability.

Figure 3:

Fig3B has a confusing construct nomenclature that needs clarification so that readers can understand quickly without having to read on for it to eventually become clear. Also remove the last 3 ZNRF3 constructs from 3B and add to 3E instead.

e.g.,

"We replaced either the extracellular domain (ECD) of RNF43 with the ECD of CD16, (CRR), its transmembrane domain (TMD) with the TMD of CD7 (RCR) as well as replacement of both (CCR) (Fig. 3B).

We adapted the sentence in and reorganized the cartoons as suggested.

What about removal to the ECD of RNF43 completely, without substitution? This may help to clarify any role it plays...

I think the following sentence needs correction "Even a chimeric RNF43 variant carrying a CD16-derived ECD along with an early truncation downstream of the RING domain (CRR323X) retained functionality in these assays, albeit with somewhat attenuated activity (Fig. 3D)."

In order to state "Thus, our results indicate that the RNF43-TMD and RING domain comprise the minimally required fraction of the protein for FZD5 downregulation", one would need to show activity for the Δ ECD/ Δ C-tail version of RNF43...

Complete removal of the RNF43 ECD leads to accumulation of the E3 ligase in the ER, likely due to misfolding of the protein (**Rebuttal figure 1**) and therefore this variant cannot be used. For these reasons, we decided to employ a domain substitution approach.

Rebuttal figure 1. Immunofluorescence showing the subcellular localisation of RNF43 WT and RNF43 deltaECD.

Referee Cross-Comments:

Both referees 2 and 3 are substantially more critical of the points that I raised regarding the data presented (especially in Figure 2). My own experience, however, would confirm that there is significant variation between the biochemistry of different FZD's and it is indeed challenging to have clearly comparable results. Nevertheless, I believe that the main conclusions of the authors are supported by the experimental data. I do however agree with some of the criticisms raised that more care needs to be taken when concluding e.g., that ZNRF3 shows specificity towards FZD6.

Overall, my opinion is that, although qualitatively sound and robust data are crucial, the "bigger picture" of a new finding is of equal significance if the data presented generally support the claims of the authors. Here, I believe this to be the case, although some improvements do indeed need to be made and more discussion on the variations between the different FZD's commented on and discussed in more detail.

More generally, in my opinion it is of more value to our field if we publish such preliminary findings more rapidly (as long as the study is sound). Often, extensive revisions may not, in the end, significantly change the overall quality or principal findings made. A slightly more tolerant approach may speed up the peer reviewing and publication process and should generally help as it will allow others to further refine the details once a new finding is published. In other disciplines, there appears to be a more forgiving/tolerant approach when it comes to peer review.

Reviewer #2 (Comments to the Authors (Required)):

The manuscript "E3 ligases RNF43 and ZNRF3 display differential specificity for endocytosis of Frizzled family members" by Madelon Maurice and co-authors aims at deciphering the substrate specificity, among the ten FZD proteins, of the two transmembrane E3 ligases RNF43 and ZNRF3. Studies on the domain(s) responsible for the substrate specificity, along with the correlations of these specificities and the relevance of the two ligases and their FZD substrates in cancer, are also provided. Overall, this is an interesting manuscript offering important and interesting insights. However, several issues question the validity of the authors' conclusions, and will need to be addressed in a major revision.

Major issues:

1. Two bands (or two groups of bands) of different FZDs visible in Westerns (Fig. 2A) are used by the authors to distinguish between the immature and mature forms of the

protein (Fig. S1)". This statement is false regarding FZD6 and FZD8, as they are not visible at all in the Fig. S1 due to the low expression levels, and the authors have to perform IPs for these receptors in their Fig. 2A. As FZD6 is a key isoform concluded in the rest of the paper to be the specific target of one of the two E3 ligases, this experimental deficit puts in question the validity of these subsequent conclusions regarding this FZD.

We agree with the reviewer this is an important point. In line with the expressed concerns, we focused on FZD6 to address this issue. We now improved the transfection and detection of FZD6, which eliminated the need for an IP step to visualize mature/immature ratios of this receptor. We repeated the experiments multiple times and show that the differential downregulation of FZD6 by RNF43 and ZNRF3 was statistically significant (**new Figure 2A and B**). In addition, due to the improved expression we were now able to visualize the plasma membrane fraction of FZD6 using cell surface biotinylation (**new Figure S1A**).

Another problem relates to FZD4: the authors acknowledge that this is the isoform for which they do not see two separate bands. This, however, does not impede the authors to present the data on the ratio of the mature to immature bands for FZD4 in Fig. 2?

We apologize for the confusion. Indeed, for FZD4 we were not able to distinguish between mature and immature forms and we therefore quantified the total FZD fraction, which we explained in the methods section. For clarity, we removed FZD4 from main figure 2B and placed the quantification of total FZD4 levels in a separate graph in the supplemental figures (**New figure S1B**).

2. The whole concept of differential specificity of RNF43 vs. ZNRF3 to the nine tested FZDs is based on the fact that the upper-migrating band of these FZDs is differentially reduced upon expression of RNF43 or ZNRF3. This is the cornerstone of the paper, and in this regard the experimental issues raised in the point 1 above regarding FZD4, FZD6 and FZD8 become even more serious. However, there are more problems with these data presented in Fig. 2. Indeed, the Westerns in Fig. 2A are representatives of only 3 experiments, as stated in the legend. In this regard, the lack of statistical significance analysis for the quantifications of these Westerns presented in Fig. 2B is very troublesome. While the data for some receptors, such as FZD5 and perhaps FZD1 and FZD7, will likely be confirmed upon the statistical analysis, the differences presented for the other receptors are two modest. And with the low number of experiments these differences won't survive, I am afraid, the statistical assessment. Yet these miniscule differences are the basis for the authors to claim that ZNRF3 has the substrate specificity towards FZD6 (already highly troublesome, as discussed above) and FZD10.

Indeed, the differences between RNF43 and ZNRF3 in downregulating FZD6 are small, but they are consistently observed across experiments. We now replaced the results of Figure 2 of the original manuscript with new results obtained using our optimized transfection and detection method, as stated above. In addition, we repeated the

microscopy experiments shown in Figure 2C-D, performed quantification of our results and added a statistical comparison of the effects of RNF43 and ZNRF3 (**new Figure 2D**). Finally, we added a FACS-based readout as a third method to quantify the effects of RNF43 and ZNRF3 on downregulating the cell surface pool of FZD6 (**new Figures 2E-F and S2C-D**). The difference in FZD6-downregulation efficiency of RNF43 and ZNRF3 was statistically significant for all three methods that we applied. Significance was not reached for the activity of both E3 ligases towards FZD10, and we thus adjusted our conclusions for this receptor.

In general, I find the way the data are presented in Fig. 2C misleading.

The reviewer raises a valid point. Since we now added ANOVA analysis to Figure 2B we think old Figure 2C becomes redundant and therefore removed it.

The data further provided in Fig. 2D, E also speak against one of the main conclusions of the authors, that ZNRF3 has the substrate specificity towards FZD6. In fact, the data cumulatively rather suggest (upon verification with the proper experimental additions and statistical assessment) that while RNF43 indeed has a preference towards some FZDs (such as FZD1, FZD5 and FZD7), ZNRF3 is unselective. Or the other way around, FZD1, FZD5 and FZD7 display the specificity towards RNF43 while the other FZDs can be more or less equally recognized by either of the two E3 ligases.

Please see our reply to point 2.

3. The interpretation of the experiments in Fig. 3A-D, where different truncation and replacement forms of RNF43 and ZNRF3 are tested for their capacity to downregulate FZD5, is made as if all the forms tested are present in equal amounts. However, these amounts are vastly different, as the anti-FLAG Western blots show, either due to their different expressions or stabilities. The data on the resulting FZD5 levels should be normalized to the levels of expression of the E3 ligase variants. Then the conclusions will be quite different from those reached by the authors. For example, the authors claim that "...replacement of RNF43-ECD with CD16-ECD (CRR) did not affect the capacity of RNF43 to downregulate FZD5..". However, CRR is expressed at the levels that look ca. 10-fold higher than the parental RNF43, thus, equal downregulations of FZD5 by the two constructs (by the way, the quantifications in Fig. 3D lack statistical significance analysis) may in reality reflect strongly decreased specific activity of CRR.

Further, the authors write that removal of the intracellular part, which was by some others reported to interact with DVL and be important for the activity of RNF43, does not affect the activity of the E3 ligase to FZD5. However, this form (called R323x) is expressed at dramatically higher levels than the parental RNF43, thus the partial reduction in FZD5 levels this form induces, when normalized to the expression levels, will essentially give loss of the specific activity of this variant.

When the authors treat their data properly, a less clear-cut picture will emerge: the TMD will indeed look crucial for the RNF43 activity towards FZD5, but the extracellular domain and the intracellular domains will also have a contributing importance, in agreement with some prior studies.

We thank the reviewer for these valuable suggestions. In our experience, E3 ligase expression displays a non-linear relation with target downregulation, however, and therefore the suggested normalization strategy would not be appropriate. Instead, we optimized expression levels of the E3 ligases and their variants, repeated the experiments, and performed statistical analysis of the results (**new Figure 3D**). Our findings confirm that CRR potently downregulates FZD5. In line with the observations of the reviewer, the S323X truncation displayed attenuated efficacy for downregulation of FZD5. Accordingly, we have adapted our conclusions, on page 7, line 208.

The same normalization to the expression levels will probably increase the strength of the authors' conclusions regarding the transmembrane domain-swapping experiments that create chimeras between RNF43 and ZNRF3. In Fig. 3E that investigates the downregulation of FZD5, the authors currently struggle with the fact that both chimeras they test (called RZR and ZRZ) have similar, intermediate levels of FZD5 downregulation. However, RZR has very high expression levels, thus its specific activity is quite reduced, while the specific activity of ZRZ will be higher given its low expression levels. The same logic then will apply to the FZD6 experiments in Fig. 3F.

As above, we repeated the experiments to achieve equal expression levels of the E3 ligases (**new Figures 3F and 3G**) and performed statistical analysis. The results confirm our previous conclusions regarding the activity of the ZRZ and RZR variants towards FZD5. For FZD6, however, TM domain swapping did not affect receptor downregulation for both RNF43 and ZNRF3. We thus conclude that the RNF43 TM domain is important for downregulation of FZD5. By contrast, we find that downregulation of FZD6 does not depend on the TM regions of either RNF43 and ZNRF3. We adapted our conclusions and added a discussion of these findings to the revised manuscript (page 8, line 260-262).

Other issues:

1. The data provided in Fig. 1 need precisions regarding statistics. While multiple repeats are behind the panel (A) as evidenced by the error bars, the figure legend does not specify the number of experiments nor what the error bars are (sd or sem).

We thank the reviewer and added the requested information to the legend.

For the panel (B), it is stated that a representative experiment of 4 independent experiments is shown. While this is fine for the Western blot shown, the quantification shown must be given as means \pm sd (or sem). Further, the statistical analysis of the significance of the differences presented must be provided (and its nature described in the legend).

We now added the requested information. We added a graph showing quantification of our results in Fig 1B, including mean, sd and p-values and we describe the applied statistical methods in the methods section.

All these points also apply to the panel (C), where even the number of experiments is not stated, so currently it can be deduced that this panel represents a single experiment.

We added a quantification of these experiments, including mean, sd and p-values to figure C.

2. The nature of error bars in Fig. 2B, C is not specified. Analysis of statistical significance in the panel (B) must be performed and provided.

We added this information to the figure legend. In addition, we performed ANOVA analysis of the data shown in Fig 2B and added the results to the figure.

3. In Fig. 3A, the lower Western blot panel is supposed to show the levels of expression of FLAG-tagged RNF43 and ZNRF3 constructs, either with the endogenous signal peptides or with the H2-Kb signal peptides. However, the bands are hardly seen. And it is not at all clear why changing the endogenous signal peptide with the exogenous H2-Kb signal peptide (both should cleave off) should produce the proteins with unequal molecular weight.

We thank the reviewer for raising this point. The constructs that were used in this experiment were differently tagged (single FLAG versus double FLAG, double HA). We now clarified this in the figure legend.

Reviewer #3 (Comments to the Authors (Required)):

The paper from Bugter and colleagues examines the role of two E3 ligases, RNF43 and ZNRF3 in the endocytosis of the Wnt receptors Frizzled.

They report that loss of function of these enzymes results in the stability of Fz receptors at the cell surface resulting in increased sensitivity to Wnt ligands. RNF43 preferentially down regulates FZD1, 5 and 7 whereas ZNRF3 affects FZD6 and FZD10. The authors went on to examine the domains of these enzymes that affects this downregulation. They also examined the tissue expression of these two enzymes in relation to where FZD receptors are expressed. The data presented seems minimal and an incomplete story. In addition, the authors did not perform statistical analyses in many of the graphs presented. This is really surprising.

The manuscript has limited amount of data and the data is poorly presented. Therefore,

this manuscript is not appropriate for Life Science Alliance.

Specific comments:

1) No statistical analyses are presented in Figure 1 B and C, Figure 2 D, E and F etc.

We now added statistical analysis to Figures 1B, 1C, 2B, 2D, 3D, 3F and 3G.

2) The figures are not properly labelled.

We improved the labeling of the figures where relevant. We changed the axis labeling in Figures 1B and 1C to more accurately inform on how the data was quantified/normalized. Secondly, by removing FZD4 from Figure 2B, the axis legend is now correctly stating 'mature/immature FZD'. We added a separate **new Figure S1B** showing total FZD4 levels. Furthermore, throughout the manuscript we improved the labelling of the western blots to make it more clear which epitope tags are stained for which constructs (e.g. in Figure 3).

3) Figure 3 C, the images are not informative without quantification.

The images presented in Figure 3C are only meant to show that E3 ligase variants display a similar localization. Expression levels are shown on Western blots shown in other panels of this figure.

June 24, 2024

RE: Life Science Alliance Manuscript #LSA-2024-02575-TR

Prof. Madelon M. Maurice
University Medical Center Utrecht
Center for Molecular Medicine and Oncode Institute
Heidelberglaan 100
Utrecht 3584CX
Netherlands

Dear Dr. Maurice,

Thank you for submitting your revised manuscript entitled "E3 ligases RNF43 and ZNRF3 display differential specificity for endocytosis of Frizzled receptors". We would be happy to publish your paper in Life Science Alliance pending final revisions necessary to meet our formatting guidelines.

- please address Reviewer 2's remaining comments
- please be sure that the authorship listing and order is correct
- please consult our manuscript preparation guidelines <https://www.life-science-alliance.org/manuscript-prep> and make sure your manuscript sections are in the correct order
- please add your main and supplementary figure legends to the main manuscript text after the references section

A. FINAL FILES:

B. MANUSCRIPT ORGANIZATION AND FORMATTING:

Thank you for your attention to these final processing requirements. Please revise and format the manuscript and upload materials within 5 days.

Sincerely,

Reviewer #1 (Comments to the Authors (Required)):

The authors have delivered a detailed revision of their manuscript, addressing all points raised to a satisfactory level. I have no outstanding concerns and recommend publication of the findings.

Reviewer #2 (Comments to the Authors (Required)):

The authors have adequately addressed my and other reviewers' comments in their major revision. I only wish them to correct / clarify the paragraph on page 7 (lines 199 to 212). In this paragraph, the authors, probably by an error appearing through rewriting the text during revision, do not clearly state that the complete removal of the C-terminus of RNF43 (cosntruct S262x) leads to the loss of FZD5-downregulating ability (although this data is present on the figure). In the paragraph it is also not clear whether the Dvl-interacting region is removed in the S323x truncant (and the S262x truncunt) or only in the S262x truncunt. So please add some more clarity to this description.

June 25, 2024

RE: Life Science Alliance Manuscript #LSA-2024-02575-TRR

Prof. Madelon M. Maurice
University Medical Center Utrecht
Center for Molecular Medicine and Oncode Institute
Heidelberglaan 100
Utrecht 3584CX
Netherlands

Dear Dr. Maurice,

Thank you for submitting your Research Article entitled "E3 ligases RNF43 and ZNRF3 display differential specificity for endocytosis of Frizzled receptors". It is a pleasure to let you know that your manuscript is now accepted for publication in Life Science Alliance. Congratulations on this interesting work.

DISTRIBUTION OF MATERIALS:

Again, congratulations on a very nice paper. I hope you found the review process to be constructive and are pleased with how the manuscript was handled editorially. We look forward to future exciting submissions from your lab.

Sincerely,
